# Identification and Antimicrobial Resistance of *Dermatophilus congolensis* from Cattle in Saint Kitts and Nevis

**DOI:** 10.3390/vetsci8070135

**Published:** 2021-07-16

**Authors:** Ian Branford, Filip Boyen, Shevaun Johnson, Samantha Zayas, Aspinas Chapwanya, Patrick Butaye, Felix N. Toka

**Affiliations:** 1Department of Biosciences, Ross University School of Veterinary Medicine, Basseterre 00334, Saint Kitts and Nevis; IBranford@rossvet.edu.kn (I.B.); ShevaunJohnson@students.rossu.edu (S.J.); SamanthaZayas@students.rossu.edu (S.Z.); pbutaye@rossvet.edu.kn (P.B.); 2Department of Pathology, Bacteriology and Avian Diseases, Faculty of Veterinary Medicine, Ghent University, 9820 Merelbeke, Belgium; filip.boyen@ugent.be; 3Department of Clinical Sciences, Ross University School of Veterinary Medicine, Basseterre 00334, Saint Kitts and Nevis; achapwanya@rossvet.edu.kn; 4Department of Preclinical Sciences, Institute of Veterinary Medicine, Warsaw University of Life Sciences—SGGW, 02-786 Warsaw, Poland

**Keywords:** *Dermatophilus congolensis*, dermatophilosis, minimal, inhibitory, concentration, susceptibility, antimicrobial, resistance

## Abstract

Dermatophilosis is a form of dermatitis caused by the bacterium *Dermatophilus congolensis*. The disease usually presents as localized purulent dermatitis, crusty hair masses or widespread matting of the hair. This condition is most common in domestic ruminants; but it can also affect other wild animals and humans. Antimicrobial therapy is used in many regions to treat clinical dermatophilosis with varying results. In this study, we aimed to assess the antimicrobial susceptibility of *D. congolensis* isolates. Fifty-two isolates were obtained from animals showing clinical signs of the disease at farms in St. Kitts. The isolates were then confirmed as *D. congolensis* by phenotypic tests, PCR and MALDI-TOF Mass Spectrometry. Furthermore, minimum inhibitory concentrations (MIC) of 16 antimicrobial agents were determined, using the broth microdilution method. Although most antimicrobials showed MICs in line with published values, the tetracycline results displayed a clear bimodal distribution over the tested range, with most isolates showing low MICs and 6 isolates much higher values (+/− 100-fold increase). These results indicate the presence of acquired tetracycline resistance in *D. congolensis* on the island of St. Kitts. Whether the current observation has implications for efficacy of treating the disease must be confirmed in further research.

## 1. Introduction

*Dermatophilus congolensis* is a facultatively anaerobic actinomycete that can infect a wide range of animals as well as humans, leading to the skin disease dermatophilosis, also commonly referred to as mycotic dermatitis (erroneously as it is not a mycosis), rain rot, rain scald or streptotrichosis [1,2,3,4]. The acute form of the disease is mainly localized in the epidermis and clinically manifests as purulent, crusted and matted hair masses formed as a direct result of the pustular process [2]. The first reported case of dermatophilosis was in Congo in 1915: the disease has since been described worldwide, although it is mainly present in areas with hot and/or humid climates. It is most often associated with cattle, sheep, goats and horses; and causes economic loss due to damaged hides, loss of body condition, poor health, secondary infections, high culling rates or, in rare cases, death [3,5].

Although rare, dermatophilosis is also a zoonotic disease, manifesting as keratolysis, pustules or exudative scaly lesions in humans [6]. It is usually seen in people who have constant contact with infected animals; however, the disease in humans has not been reported to have any systemic complications such as those seen in animals [6]. In 1961 the first four cases of this disease were reported in humans in the United States, with all four individuals having prior contact with infected deer [7]. There is also evidence that this microorganism is associated with some forms of pitted keratolysis in humans [8]. Due to the diagnostic procedures not being well-developed, it is likely that this condition is underdiagnosed in humans [5].

*D. congolensis* is spread through direct contact with infected animals, insects or fomites and the disease is proliferated by continuous rainfall, humidity and heat [6]. The prevention of this disease is mainly achieved through management factors such as separation of the animals, sheltering the animals and protection from too much moisture if possible. In cases where the infection is widespread, treatment is necessary and antimicrobials and acaricides are used [3,4,9]. Mainly penicillin, aminoglycosides and tetracyclines are used to treat infected animals [4,9]. Acquired antimicrobial resistance has not been reported to date, even though antimicrobial treatments are common. However, there are few reports of comprehensive minimal inhibitory concentration (MIC) and Minimal bactericidal concentration (MBC) data [6]. Given the large problems with infections caused by *D. congolensis*, we obtained isolates from clinically affected animals, assessed the use of MALDI-TOF in the identification of *D. congolensis* and investigated *D. congolensis* isolates for their antimicrobial susceptibility to determine the epidemiological cut offs (ECOFFs) and potential acquired resistance.

## 2. Materials and Methods

### 2.1. Sample Collection

Cattle (n = 47) from 10 farms in St Kitts were examined from February 2019 to February 2020. Animals sampled were predominantly adults and female but included other cattle with evident disease regardless of age or sex. In total 85 samples were collected of which 47 were scabs and 38 swabs. The samples were placed in a labeled test tube and transported to the laboratory at room temperature within 1 h of collection. This study did not involve human subjects but was approved for animals by the Institutional Animal Care and Use Committee (IACUC) at Ross University School of Veterinary Medicine, #21.03.06Toka.

### 2.2. D. congolensis: Isolation and Identification

In the laboratory, swabs were inoculated directly onto TSA (trypticase soy agar) w/5% Sheep Blood (REMEL Inc., LEXENA, KS, USA), while Haalstra’s method [10] was used for the primary isolation of *D. congolensis* from scab samples. Briefly, for Haalstra’s method a small amount of the scab was ground up and placed in a tube containing 2 mL of distilled water for 3 h at room temperature. The suspension was then placed in a jar with a lit candle at room temperature for 15 min to concentrate motile zoospores of *D. congolensis* to the top of the suspension. Next, a loopful was inoculated onto a TSA w/5% Sheep Blood plate (REMEL Inc., LEXENA, KS, USA). All plates were incubated for 48 h at 37 °C in a 5% CO_2_ enriched atmosphere. After purification, preliminary identification of *D. congolensis* was performed based on the phenotypic appearance and classical biochemical reactions of indole and catalase. Identification was confirmed by PCR, as described [11] with slight modifications. Genomic bacterial DNA was purified using the Qiagen Allprep bacterial DNA/RNA/protein kit (Qiagen, Hilden, Germany. The *agac* gene that encodes alkaline ceramide protein was amplified with the primers forward:5′-TGGCAGCTCTGATGAGTACCACAA-3′; Reverse:5′-AATG-TGCCGGGAACGGAAATCAAC-3′ to produce a 127 bp product [4]. We also assessed the capacity of MALDI-TOF MS (Brucker Daltonics, Germany), for the confirmation of *D. congolensis*. A single colony was transferred onto a polished steel plate, air dried and covered with 1 μL alpha-cyano-4-hydroxycinnamic acid matrix (Bruker Daltonics, Bremen, Germany). The sample was identified with an Autoflex III smartbeam MALDI-TOF MS, using FlexControl and MBT Compass software (Bruker Daltonics, Bremen, Germany). Log score values higher than 2.00, between 1.70 and 1.99, and between 0 and 1.69, indicate high-confidence, low-confidence, and no-confidence identification, respectively, as general guideline applied by the manufacturer. In case the direct transfer method did not result in a score value >1.69 for a specific isolate, the extended direct transfer method, including an extra formic acid extraction step was used for these isolates.

### 2.3. Susceptibility Testing

Susceptibility testing was performed using the broth microdilution method with Mueller Hinton II broth (REMEL, LEXENA, KS) according to the CLSI guidelines (CLSI VET01 and CLSI VET08) [12,13]. Broths were incubated at 37 °C and 5% CO_2_ for 72 h. The control strains were *Escherichia coli* ATCC 25922, *Pseudomonas aeruginosa* ATCC 27853, *Staphylococcus aureus* ATCC 29213 and *Enterococcus faecalis* ATCC 29212 as well as the reference strain *D. congolensis* ATCC 14637. All isolates were tested in duplicate with the higher concentration taken as the final in cases when variations occurred. Concentrations tested ranged between 0.125 µg/mL and 64 µg/mL for all antimicrobials. Susceptibility was tested against chlortetracycline, ampicillin, florfenicol, tetracycline, sulfadimethoxine, tylosin, novobiocin, neomycin, amoxicillin, danofloxacin, enrofloxacin, ceftiofur, bacitracin, trimethoprim, penicillin and tulathromycin A (all acquired from SIGMA-ALDRICH, Saint Louis, MO, USA). Differentiation between susceptibility and resistance was performed according to the epidemiological cut off (ECOFF) criteria, or otherwise named the microbiological criterium, using visual estimation (“eye-ball” method) [14].

## 3. Results

### 3.1. D. congolensis Isolation and Identification

Of the 85 specimens collected we recovered 52 isolates of *D. congolensis*. The percentage of isolation from swabs was 52% and 68.1 from scabs. Both scabs and swabs were collected to increase of isolation of the bacteria. Swabs were also collected from only a few healthy animals without lesions. Samples from such animals were negative of *D. congolensis* and data are not included. Of these 52 isolates we took 44 random samples for the confirmatory testing; the isolates were all from different animals. All isolates were identified as *D. congolensis* using MALDI-TOF MS with score values >1.69 (Appendix A), mostly using the extended direct transfer method. Although for 17 isolates, high-confidence score values (x > 1.99) were obtained, for the remaining isolates low confidence score values (1.99 > x > 1.69) were obtained. The results obtained by PCR (Figure 1) confirmed the MALDI-TOF results.

### 3.2. Susceptibility of D. congolensis

All the 52 isolates were subjected to antimicrobial susceptibility testing, but the results summarized in Table 1 show only 34 isolates reflecting a single isolate from each animal. Quality control strains tested had results that fell within the recommended CLSI provided ranges; however, there are no available ranges for the *D. congolensis* ATCC 14637 reference strain. The susceptibility of the isolates to the antimicrobial chlortetracycline ranged from 0.25 µg/mL to 64 µg/mL. Tetracycline has the strains distributed over MIC ranges from ≤0.125 µg/mL to 32 µg/mL. The isolates showed relatively high results over a wide range of concentrations against sulfadimethoxine, with results between 4 µg/mL and >64 µg/mL and just one isolate being inhibited at 0.5 µg/mL. Trimethoprim MIC distribution was between 0.5 µg/mL to 4 µg/mL while both of the fluroquinolones, danofloxacin and enrofloxacin had a very narrow MIC range between 2 µg/mL to 4 µg/mL and 2 µg/mL, respectively. For the penicillin we observed a MIC distribution between 0.25 µg/mL and 1 µg/mL for ampicillin, between ≤0.125 µg/mL and 0.5 µg/mL for penicillin and between 0.5 µg/mL to 1 µg/mL for amoxicillin. Tylosin and tulathromycin A had a MIC of 0.25 µg/mL against all isolates. Neomycin had a MIC of 2 µg/mL against all isolates. Ceftiofur showed a MIC range of 0.25 µg/mL to 1 µg/mL, florfenicol from 0.5 µg/mL to 1 µg/mL. Novobiocin results were between 1 µg/mL to 2 µg/mL and bacitracin results were between 0.5 µg/mL to 2 µg/mL. The MICs of antimicrobials against the *D. congolensis* ATCC strain (marked by * in the table) are comparable to the MICs of the isolated strains.

Using the ECOFF criteria, whereby the population is assessed for multimodality (https://eucast.org/mic_distributions_and_ecoffs/ accessed 5 May 2021), we found a bimodal distribution of the strains for the tetracyclines chlortetracycline and tetracycline. For chlortetracycline the susceptible MICs ranged between 0.25 µg/mL and 0.5 µg/mL while the resistant MICs were between 32 µg/mL and 64 µg/mL. In the case of tetracycline, MICs defined as susceptible were ≤0.125 µg/mL while the resistant MICs ranged from 16 µg/mL to 32 µg/mL. The ECOFF for the sulfonamide sulfadimethoxine is rather hard to determine, as only one strain had an MIC of 0.5 µg/mL, while the others had MICs of ≥4 µg/mL.

## 4. Discussion

*D. congolensis* is a major problem in some cattle-rearing regions of the world, including the Caribbean. Many infected animals usually develop severe clinical symptoms suggesting ineffective treatment. Here, we aimed to assess whether MALD-TOF is a good identification method for *D. congolensis* and we performed MIC testing on isolates from cattle on St Kitts to determine antimicrobial susceptibility. In our experience, scab material showed the best isolation rates compared to swabs.

MALDI-TOF is a fast, analytical method; however, the equipment is rather expensive and, as such, a high throughput is necessary for cost-effectiveness. Many clinical laboratories rely on MALDI-TOF for the identification of bacteria but only a few veterinary diagnostic laboratories employ this method. In this study we confirm that MALDI-TOF is an excellent tool for the identification of *D. congolensis,* even though for some isolates several attempts were needed, including an additional extraction step with formic acid. Although for more than half of the isolates low confidence score values were obtained, the use of a score value > 1.69 for reliable identification of, for example, *Staphylococcus* species has been described before [15] and may also be suggested for the identification of *D. congolensis*, even though this should ideally be confirmed in a larger trial. In addition, both for high and low confidence score results, no other bacterial species were suggested with score values > 1.69, making the interpretation straightforward.

There are no breakpoints for *D. congolensis*. The basis for interpretation of the MICs in our study was thus based on the microbiological criterion, using the Epidemiological cutoff (ECOFF) as defined by European Committee on Antimicrobial Susceptibility Testing (EUCAST, https://eucast.org/mic_distributions_and_ecoffs/, accessed 5 May 2021). We cannot assess the pharmacological criterion, which considers the serum levels of the antimicrobial, nor the clinical criterion, due to the lack of clinical data. The clinical efficacy cannot be determined by the MICs only. However, clinical studies have indicated the efficacy of several antimicrobial treatments. Tetracyclines, long acting oxytetracycline in particular, have been shown to be effective in the treatment of dermatophilosis [16,17]. However, using the microbiological criterium, we demonstrated that six isolates had higher susceptibility results for chlortetracycline and tetracycline, strongly indicating that these isolates had an acquired resistance mechanism. The microbiological criterion refers to direct in vitro interactions between the antimicrobial agents and the *D. congolensis* isolates and does not necessarily predict how the patient will respond to therapy. However, tetracycline MIC values were at least 10 to 100 times higher for isolates with potential acquired resistance. The likelihood that animals infected with these isolates would respond well to treatment is low.

Penicillin-streptomycin, penicillin alone, as well as amoxicillin and lincomycin-spectinomycin have also been reported to successfully treat dermatophilosis [3,4,18], although the condition was not completely cured in all cases. Suboptimal results were achieved when erythromycin was tested in sheep with dermatophilosis [18]. Our study showed very low MICs for both penicillin and Amoxicillin and a monomodal distribution of results, indicating these antimicrobials indeed have the potential to be good therapeutics.

It should be mentioned here that this study did not focus on defining the prevalence of dermatophilosis in cattle but on collecting *D. congolensis* and investigating its antimicrobial susceptibility. Few studies have assessed the normal antimicrobial susceptibility of *D. congolensis* and the most detailed one is at least 25 years old; therefore, the current results provide additional insights in both intrinsic susceptibility and potential acquired resistance against various antimicrobial agents. The currently described isolates are, however, obtained from one location, while a much older report assessed a panel of strains from international locations including the USA, Spain, parts of Africa, Japan and the Caribbean [9]. Another difference was that the previous study used a pharmacological based breakpoint (based on the concentrations achievable in serum), and resistance was reported based on these assumed breakpoints [9]. However, the data did not show multimodality in the distribution of the strains over the MICs of the antibiotics, indicative of acquired resistance. Resistance against nitrofurans was reported as the highest, with 87.5% of isolates tested exhibiting resistance against furaltadone and all isolates showing resistance against nitrofurazone, suggesting these antimicrobials are not suitable for treatments [9]. The isolates tested against the sulfonamides also showed a high prevalence of resistance (42.1%) while resistance against the aminoglycosides, gentamicin and neomicin, were 15.8% and 5.26%, respectively [9]. Additionally, for the antimicrobials, there were no indications of acquired resistance as the strains distributed monomodally over a narrow range of MICs as in our study, except for sulphonamides. Sulphonamide susceptibility is difficult to assess due to the wide range of MICs, and it is unclear in this case what this wide range means. 

This is the first description of high tetracycline resistance in *D. congolensis*. The distribution of tetracyclines is clearly a bimodal distribution of the strains over the MICs, with one group showing lower and another group higher MICs. It is unclear whether this higher MIC also represents clinical resistance; however, it can be expected that this will negatively affect the clinical outcome of a tetracycline treatment. The genetic background of this resistance needs further investigation. In St Kitts the main treatment is long-acting oxytetracycline (personal communication: Shevaun Johnson). The use of this antibiotic in the past may have selected for resistance in these strains and future use of this antimicrobial may be compromised. This is also evident in the history of significant losses and severe disease of cattle reported on farms in St Kitts.

CO_2_ incubation affects the MICs of several antimicrobials such as macrolides [19] and aminoglycosides [20] and because of this the true values, useful for assessing the potential clinical efficacy, should potentially be taken lower. Several bacteria, such as *Legionella* spp., *Streptococcus* spp., *Moraxella* spp. and *Haemophilus* spp., also have to be incubated in the presence of CO_2_, otherwise they would not grow, or their growth would be suboptimal. Antimicrobials are only active in metabolically active bacteria [19,21,22,23]. Clearly, *D. congolensis* also needs to be incubated in a CO_2_ enriched atmosphere and this may have affected the absolute values of the MICs but did not affect the detection of potentially acquired resistance, as demonstrated for other bacteria.

## 5. Conclusions

Here, we show that MALDI-TOF is a good tool for the rapid and cheap identification of *D. congolensis*. We report on the normal susceptibilities of *D. congolensis*. The ranges obtained are in line with the available literature. We report for the first time potential acquired antimicrobial resistance against tetracycline in this bacterium. Further studies are warranted to decipher which gene and what mobile genetic elements are involved in this resistance. This will allow us to estimate the potential impact of this resistance as well as advise veterinarians on the way to mitigate the problem.

## Figures and Tables

**Figure 1 vetsci-08-00135-f001:**
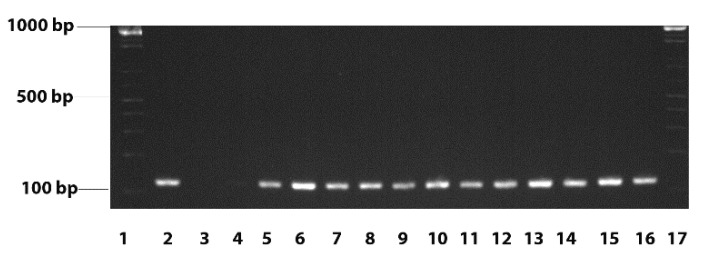
Representative PCR results of the amplification of the *agaC* gene of the *D. congolensis* isolates. Lanes 1 and 17—Molecular weight markers; Lane 2—positive control (*D. congolensis* ATCC: 14637); Lane 3—negative controls (no template DNA); Lane 4—negative control (no Taq polymerase) Lanes 5–16—*D. congolensis* isolates.

**Table 1 vetsci-08-00135-t001:** MIC result distributions of 16 antimicrobials against 34 isolates of *D. congolensis*. Results of the *D. congolensis* ATCC 14637 reference strain are indicated with an asterisk (*). Blank table cells indicate that none of the isolates had MIC values corresponding to the given concentration.

Antimicrobial	Number of Isolates with MIC in µg/mL										
	≤0.125	0.25	0.5	1	2	4	8	16	32	64	>64
Chloretetracycline		7	20	1*					2	4	
Tetracycline	28 *							3	3		
Sulfadimethoxine			1			2	3	7 *	11	5	5
Trimethoprim			3	14		17 *					
Danofloxacin					24	10 *					
Enrofloxacin					34 *						
Ampicillin		4	30 *								
Amoxicillin		1 *	15	18							
Penicillin	3 *	4	27								
Tylosin	1 *	33									
Tulathromycin	1 *	33									
Neomycin			1 *		33						
Ceftiofur	1 *	1	15	17							
Florfenicol			5 *	29							
Novobiocin				15	19 *						
Bacitracin			2	31 *	1						

## Data Availability

Data is contained within the article or Appendix A.

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
