# Peer review of "Identification and Antimicrobial Resistance of Dermatophilus congolensis from Cattle in Saint Kitts and Nevis"

_vetsci, 2021, doi:10.3390/vetsci8070135_

Round 1

Reviewer 1 Report

The Authors report a very interesting diagnostic investigation regarding Dermatophilus congolensis-associated cases in ruminants. The paper appears well written and the experimental study is appropriate and complete. Anyway, I suggest to improve the description of clinical cases in ruminants, reporting more details about number, age and sex of affected animals, type of lesions observed. In addition, the diagnostic design should be more effective with other microbiological investigations, in order to rule out other potential pathogens involved in clinical cases. 

I suggest to perform a molecular screening for antimicrobial resistance genes (ARG) associated to the acquired tetracycline resistance, to confirm or not any plasmid-mediated transfer of resistance.

Finally, if available, some information about human cases of dermatophitosis in farm workers/owners could be useful to improve the scientific interest of the paper.

Author Response

We thank Reviewer #1 for his efforts and valuable time to review our manuscript. We have attached a point-by-point response to the Reviewer's comments and suggestions. We are grateful that the incorporated suggestions have improved our manuscript. 

Regards

Reviewer 2 Report

Manuscript title:

General comments: The manuscript describes isolation and antibiotic susceptibility of

 Dermatophilus congolensisisolated from cattle. The methods are appropriate, and the manuscript is well-written. The only major item to address is animal welfare oversight by Ross University or another body in St. Kitts where the samples were collected. There are also a few minor points that should be addressed. 

Major specific comments:

  1. Please include an IACUC statement. From the instructions to authors: please add the Institutional Review Board Statement and approval number for studies involving humans or animals. Please note that the Editorial Office might ask you for further information. Please add “The study was conducted according to the guidelines of the Declaration of Helsinki, and approved by the Institutional Review Board (or Ethics Committee) of NAME OF INSTITUTE (protocol code XXX and date of approval).” OR “Ethical review and approval were waived for this study, due to REASON (please provide a detailed justification).”

Minor comments:

  1. Figure 1.Please state the gene that was amplified in the figure legend. Agac, alkaline ceramide?

  1. Table 2.I generally dislike leaving table cells blank because the reader might wonder if the blank concentration was performed.  However, the table is much easier to skim than if all the blanks contained the numeral, 0.  I recommend leaving the table as it is, but add a sentence like this to the figure legend: “Blank table cells indicate that none of the isolates had MIC values corresponding to the given concentration.”

  1. There is an indentation error in the reference list.

  1. The reviewer is uncertain about the reference style used for references 11 and 12.Please confer with the production editor.

Author Response

We are thankful for the Reviewers time and effort put into reviewing our manuscript. We have attached a point-by-point response to the Reviewer's comments and suggestions. We appreciate that the incorporated changes have improved our manuscript.

Reviewer 3 Report

Brandford et al., describes the results of the D. congolensis surveillance study in cattle in St. Kitts. The authors collected 83 skin samples (47 scabs and 36 swabs) from 47 cattle from a total of 10 farms. A total of 52 isolates of D. congolensis were recovered from 83 samples. All isolates were identified as D. congolensis by MALDI-TOFF and PCR and screened for antimicrobial resistance against several clinically relevant antibiotics. In general, the data should be of interest to the veterinary community and is relevant to this journal. The strength of this study is that the authors performed culture and identification of isolates and performed MIC testing. Given that this is a targeted surveillance study for D. congolensis in cattle, it would make more sense to clearly state the surveillance as a study objective. Given the sampling scheme and the study population presented in the methods section, it appears that more than one isolates were recovered from individual animals, however, these are presented as separate isolates (Table 1 and elsewhere). Similarly, it appears that more than one isolates were recovered from the same farm, however, these are also presented as separate isolates. The isolates from the same cattle are very likely identical and isolate from the same farm are also likely identical (the disease is contagious). Thus, isolates per cattle should be reported as such as “one” isolate. The data on the number of animals sampled/farm and the number of isolates recovered per farm should be included in a table. In the absence of this information, the total number of isolates presented appears misleading from the epidemiological standpoint. Also, repeated testing of multiple isolates from the same animal or multiple samples collected from the same animal appears to inflate the AMR data (Table 2, lines 175-177 and elsewhere in the manuscript). It would be more appropriate and informative to report AMR data on a single isolate from each animal concerning the farm of origin and revise the numbers accordingly. If authors have data on percent isolates recovered from swabs vs scab, that also needs to be reported here. I also have additional comments below and hope that the authors will find these useful.

Line 13: Dermatophilosis can present as asymptomatic to mild to moderate to severe disease. It is not always severe unless there is a significant predisposing factor. Please consider deleting the word “severe”.

Line 25 (and lines 175-177, 210, 225): The MIC data for tetracycline confirms resistance, however, it doesn’t demonstrate that the resistance is acquired or induced unless the genes (and mechanism) encoding resistance in these isolates is investigated. Suggest deleting the word “acquired”.

Line 34: please either delete the word “mycotic” or indicate that “it is erroneously referred as ‘mycotic’ dermatitis”

Line 35: “clinically manifests as….”

Line 38: please delete the word, “restricted”. The disease is common in a hot and humid climate, but it is not “restricted” to such a climate.

Line 41: It is important to recognize that “the disease rarely results in death, mostly due to the secondary complications” please revise accordingly

Line 42: Dermatophilosis is “rarely zoonotic” please revise

Line 48-49: It is important to mention here that the primary form of treatment is by management changes including moving animals to a sheltered location to protect from moisture (eg., rain) and keeping them dry. In cases where the infection is widespread, antibiotic treatment is recommended. Please revise accordingly.

Line 50: What do the authors mean by “true acquired antimicrobial resistance”?

Line 58-61: Please clarify the specific criteria used for sampling 47 cattle. Why were these animals chosen for sampling? Was this based on clinical signs? If so, were there any control animals? Also please clarify why scabs and swabs were collected simultaneously and why fewer swabs were collected than scabs? Direct smear examination of the clinical material is often used for the diagnosis of dermatophilosis. Please clarify if smears were made for direct straining of the clinical material? If so, include this information as it is clinically relevant and can be useful for practicing clinicians.

Line 63: What is TSA? Please define this at its first occurrence.

Line 75: Please clarify what agaC gene was chosen as a target for PCR? Is this gene species-specific? Also, please clarify what positive and negative control was used for PCR.

Line 90-104: It looks like the antibiotic susceptibility test was conducted under anaerobic conditions. Aminoglycosides do not work effectively under anaerobic conditions. The MIC of aminoglycosides often increases significantly under anaerobic conditions. There are several publications on this topic, here is one (https://pubmed.ncbi.nlm.nih.gov/3888545/). This is a significant caveat in this study; thus the reported MIC of neomycin (line 132) are likely very inflated and should be removed from the data.

Table 1: The information presented in table 1 can be included as supplementary material.

Line 148: I am not so sure if dermatophilosis is a major problem in cattle-rearing regions of the world. What percent of derm diseases of cattle is due to dermatophilosis?

Line 150-152: The objective of the study stated here is different from what is stated in lines 53-55. Please clearly state the objective which appears to be targeted surveillance for D. congolensis in cattle in St. Kitts. The strength of this study is that the authors performed culture and confirmed isolates via MALDI-TOFF and PCR all of which supports the disease prevalence in St. Kitts. However, focusing the discussion on diagnostic methods is a bit distracting, especially when the most common (and effective) method to detect D. congolensis is a direct examination of a gram stained smear of the scab material which often reveals typical diplococci in chains arranged in “tram track like arrangement”.  The PCR test is also rarely used. Culture is used sometimes as a follow-up. MALDI TOFF equipment is expensive however it is the most rapid method to detect bacteria and saves time, labor, and resources. So, in the long run, this is the most economic and environmentally friendly method for bacterial identification and is becoming gold standard. It would be more relevant (and appropriate) to keep the discussion focused on the prevalence of dermatophilosis in cattle in St. Kitts rather than tools used to detect D. congolensis.

Author Response

We thank Reviewer #3 for his time and effort in reviewing our manuscript, in particular, for providing us with a constructive critique of our work. We have provided a point-by-point response to the Reviewer's comments and suggestions. In our opinion the suggested changes incorporated into the manuscript have improved the manuscript.

Round 2

Reviewer 3 Report

Authors have addressed most of the comments appropriately and improved the quality of the manuscript. However a few important points were not adequately addressed in the revision.

  1. Although authors indicate that this was not the surveillance study and that this was a diagnostic investigation. This does not take away the fact that multiple isolates from the same animals inflate the results of the MIC testing. The authors could clearly describe how many redundant isolates were tested and also highlight these in suppl. Table 1 with a foot note clearly indicating that these isolates were from the same animal. Additionally, authors need to revise Table 1 in the manuscript to only show data for 1 isolate per animal and could identify in foot note if there were multiple isolates tested in some cases with  a note that resistance was similar/different.
  2. The authors explain in their responses that they realize that the testing conditions for antibiotics are not ideal, however they don't discuss this in the MS. It is important to discuss this caveat for neomycin and macrolides. 

Author Response

The authors greatly thank Reviewer #3 for additional suggestions that we think have indeed made the manuscript better. We hope that the manuscript can be further considered for publication in MDPI Veterinary Sciences.

Regards
